# Ozone as Redox Bioregulator in Preventive Medicine: The Molecular and Pharmacological Basis of the Low-Dose Ozone Concept—A Review

**DOI:** 10.3390/ijms242115747

**Published:** 2023-10-30

**Authors:** Renate Viebahn-Haensler, Olga Sonia León Fernández

**Affiliations:** 1Medical Society for the Use of Ozone in Prevention and Therapy, D-76473 Iffezheim, Germany; 2Pharmacy and Food Institute, University of Havana, Calle 222 # 2317 e/23 y 31, Havana 10 400, Cuba

**Keywords:** COVID-19, innate immune memory, rheumatoid arthritis, autoimmune disease, redox bioregulation

## Abstract

The best form of prevention against human infection through bacteria, viruses, and other parasites is ozone disinfection of wastewater and drinking water as a highly effective, well-known method. Various preclinical studies showed promising results, which are being revisited and reconsidered in times of pandemics and led to interesting results in recent clinical trials and reports, as presented by the example of protective measures against COVID-19 in particularly vulnerable clinical personnel. The application of ozone in the form of the low-dose concept induces its regulation by interference of ozone or its peroxides into the redox equilibrium of the biological system, which finally results in the restoration of the glutathione equilibrium. The antioxidant system is activated, the immune system is modulated, and thus the defense mechanisms are improved. In patients with rheumatoid arthritis, repeated ozone treatments have led to new findings in “immunomodulation” through ozone. The more effective immune response is discussed as the response of innate immune memory and opens interesting aspects for complementary treatment of autoimmune diseases.

## 1. Introduction

Times of pandemics are times when health maintenance, protection against infections, prevention in general, and protective measures come to the fore. At the turn of the 19th/20th century, sewage and drinking water were recognized as one of the major epidemic causes of typhoid, cholera etc., and appropriate water treatment plants were built. This nightmare soon became a thing of the past, unfortunately, not everywhere in the world even up to the present. Ozone as one of the most potent disinfectants is installed to make use of its high oxidation potential for the degradation of toxic substances; its germicidal, bactericidal, and antiviral properties guarantee perfect drinking water, the most important basic food [1].

### 1.1. Prevention through Room and Surface Disinfection

To protect against germ transmission in clinics and hospitals, various surface and room disinfectants are used that must be registered and listed as disinfectants following both national and international regulations. As a gas, ozone occupies a special position. Despite its pronounced bactericidal and virus-inactivating character, it requires a high humidity of >80% and is therefore, due to its high redox potential, not very compatible with materials. In addition, its toxicity to the respiratory tract requires significant measures to completely decompose the ozone before rooms can be used again [2].

In contrast, in wastewater, drinking water, and industrial water, ozone is superior to all known disinfectants, depending on concentration and contact time (concentration-time concept), well documented for many decades [3].

Ozonated water can be used in the medical field for surface and equipment disinfection, of particular benefit in dentistry, since viruses such as corona are also inactivated [4,5,6,7,8].

The underlying molecular mechanism is mainly based on direct contact with the germs; at pH values > 7, the radical mechanism dominates [9].

Of interest in medicine, this mechanism is employed in the local treatment of infected wounds for effective elimination of microorganisms and wound cleansing as a precondition for wound healing. In the systemic application and action of ozone, we discuss an indirect mechanism via ozone peroxides (Figure 1); this is described in Section 5 and in detail in [10].

### 1.2. From Empirical Findings to Science

In medical applications, ozone occupies a subordinate place in prevention although, in many private practices and clinics, the systemic use of ozone has shown remarkable success, especially in the secondary prevention of chronic diseases. In the 1990s, Schulz et al. [11] transferred these reports of experience into an animal model of lethal peritonitis and used preventive intraperitoneal ozone injections to increase the survival rate of animals up to 100%, depending on their combination with antibiotics. Systematic studies using animal and cell models were in an initial phase, and the activation of cell metabolism, immunomodulation, and redox regulation were recognized as the underlying biochemical and pharmacological mechanisms. Prevention of liver and kidney intoxication by chemotherapeutic agents or radiation exposure through ozone oxidative preconditioning are well known and have been discussed earlier [12]. After two introductory historical preclinical studies providing us with a basic understanding, we here focus on very new aspects such as protection against infections including COVID-19 and secondary prevention in rheumatoid arthritis (RA) by claiming involvement of the immune memory. Table 1 and Table 2 list the preclinical and clinical trials on the preventive effect of ozone without claim to completeness.

## 2. Preclinical Trials in Animal and Cell Models

### 2.1. Growth Inhibition of Plasmodium Falciparum in Ozone-Pretreated Red Blood Cells (RBC)

Plasmodium falciparum, the single-celled parasite inducing malaria tropica, grows in red blood cells (RBC’s) after passing most of its life cycle up to the RBCs’ bursting and consequent release of the parasites into the blood plasma. The sensitivity of plasmodia to reactive oxygen species is well known and is the basis of various drugs, such as artemisinin, an endoperoxide from the group of sesquiterpenes [25]. Thus, the idea of using ozone as an oxidant in prevention and/or therapy as a drug was obvious. The therapeutic method failed completely, at least with a single ozone treatment of infected RBCs whereas one single pretreatment with ozone prior to infection inhibited plasmodium growth by a factor of up to seven [26].

At the high concentration of 80 µg/mL, ozone appears to dramatically reduce the antioxidative capacity of RBCs by disturbing the GSH ⇌ GSSG balance and shifting it to the right (GSH is a reduced form of glutathione and GSSG is the oxidated form). Red blood cells are constantly exposed to reactive oxygen compounds and therefore have a high level (90%) of the most important cellular antioxidant GSH; this is crucial for survival and growth of plasmodia. Similar effects can be seen in patients with a G-6PDH (glucose-6-phosphate dehydrogenase) deficiency, the enzyme that initiates the pentose phosphate pathway with its antioxidant ability: these patients are mostly malaria resistant, see Figure 2.

RBCs have no nucleus and cannot initiate a repair mechanism like other cells; the GSH/GSSG balance is disturbed at high ozone concentrations (here 80 μg/mL), detoxification of peroxides no longer occurs, and the antioxidant protection of the cell is lost. This is one of the main reasons for not exceeding ozone concentrations of about 40 µg/mL during systemic ozone treatment.

### 2.2. Protection against Lethal Peritonitis by Oxidative Preconditioning with Ozone in Animal Models

With our knowledge of the immunomodulating effect of medically used ozone [28] and the idea of establishing protection against bacterial infections in both humans and animals through preventive ozone administration [11,29], Schulz et al. determined the survival rate in a suitable animal model (rats) of lethal peritonitis as a function of intraperitoneally administered ozone concentrations (10 to 100 μg/mL) and dose. Here, lethality was remarkably reduced from 95% (control) to 35%, and interestingly, at the lowest concentration of 10 μg/mL (20 mL) corresponding to a dose of 200 μg, administered once daily for 5 consecutive days (Figure 3). Synergistic effects with different antibiotics in the follow-up studies increased the survival rate to 93% and 100% in the case of Tazobac/Piperacillin TZP, accompanied by a reduced amount of IL-1β and TNF-α mRNA in spleen and liver [30,31]. 

## 3. Low-Dose Ozone Concept in Preventive Medicine

These early preclinical studies on prevention provided us with important guidance on dosing and establishing the low-dose ozone concept.

In contrast to water disinfection, we follow the low-dose concept in medicine: For wound disinfection, we use concentrations in the range of 70 to 100 μg/mL, for systemic applications, 10 to 40 μg/mL and volumes of 50 to 100 mL in the case of major auto hemotherapy (MAH), and 100 to 300 mL when using rectal insufflation (RI). Table 3 shows a general overview and detailed information is given in [10,32].

Redox bioregulation through oxidative preconditioning by ozone at low concentrations is the basic mechanism of ozone in prevention. In addition to immunomodulation, oxidative protection mechanisms by upregulation of antioxidative enzymes, such as GSHox, GSHred, SOD, CAT, and others, dominate in prevention. Healthy cells with a strong antioxidant capacity are largely protected against reactive oxygen species and free radicals generated by infection, inflammation, antibiotics, or other drugs. In the late 1990s, León started a broad-based research project on ozone-oxidative preconditioning, that rapidly expanded to attain worldwide acceptance: protection through ozone, especially against liver and kidney intoxication through xenobiotics, remedies, chemotherapy, or reperfusion damage in surgery [33,34,35,36].

A variety of animal models and a few clinical trials with excellent, convincing results form the basis for our molecular and pharmacological understanding of the use of ozone in preventive medicine. By its very nature, it is difficult to convince healthy people to take preventive measures. Although awareness, interest, and demand for prevention are currently increasing in clinics and hospitals, studies on prevention are rare.

In recent decades, cellular and animal models have been developed and used to understand empirical results in the application of ozone, to clarify the underlying mechanisms and to establish guidelines, and the idea of prevention has emerged. Although there are several thousand scientific publications on the use of ozone in biology and medicine, prevention is rarely found in the corresponding medical databases. We therefore searched the extensive, relevant literature that we were aware of to meet the increasing demand for preventive measures in the post-pandemic period, presently subject to heated discussions. Table 2 and Table 3 provide a good overview, and a few of them are discussed here, especially those that open and have opened new aspects.

Clinical studies and reports involving elderly patients [15] in the prevention of COVID-19, the corona-induced disease, show very promising outcomes. Table 2 lists these. We will now discuss some of the most interesting results.

## 4. Clinical Trials and Reports

### 4.1. Prevention against COVID-19

The best remedy for viral diseases is prevention, and the question is: Can we contribute with low-dose, systemically administered ozone in the form of major auto-hemotherapy MAH or RI and consequently protect particularly vulnerable individuals against COVID-19?

Physicians and clinical staff caring for COVID-19 patients find themselves in such a difficult situation and are more likely to be open to preventive measures. We therefore focus on clinical reports involving this group.

A. A clinical trial conducted at the Indonesian Police Central Bhayangkara Hospital and the Department of Forensic Medicine, Jakarta University, presented at the 26th Ozone World Congress of the International Ozone Association IOA in Milan, Italy, July 2023.

The trial consisted of 150 healthcare professionals, including 25 physicians and 125 nurses (exclusion G-6PDH glucose-6-phosphate dehydrogenase deficiency), who received systemic ozone treatments in the form of MAH, RI, or vaginal insufflation on five consecutive days before coming into contact with the COVID-19 patients in the hospital with a high risk of infection via the coronavirus. Concentration and dose were chosen according to the guidelines [32]. None of the 150 healthcare professionals were infected by the coronavirus, all of them remained PCR negative. Preventive ozone treatment was able to stabilize the immune status of the involved persons. At the same time, there was a nursing emergency in other hospitals due to the high sickness rate, and care of the patients collapsed [21].

B. A retrospective study at the Marmara University together with the health sciences University Sisli, Istanbul, reports the very same results: Could ozone therapy be used to prevent COVID-19?

This retrospective study consisted of 71 persons who completed at least 10 sessions of MAH in accordance with the low-dose ozone concept within six months; 45% of them were medical professionals, all of them had contacts and travel history; 50.7% were aged over 50 years and 52% had comorbidities. Consequently, their risk of developing COVID-19 was higher than that of the normal population in Turkey [22]. Two of the included persons were infected, acquiring COVID-19 without severe symptoms corresponding to 2.8% of the participants.

C. The effect of systemic ozone application on the activity of B-lymphocytes and the level of antibodies in corona-vaccinated persons.

A retrospective clinical trial (Goji Medical Group Ozone Therapy Institute, Miami Lakes, FL, USA) including 109 persons, mostly triple vaccinated, 57 of them receiving systemic ozone application as MAH (60%) or RI (40%); 52 persons not treated with ozone served as controls (Table 4).

No person in the ozone group was PCR positive compared to 48 persons in the controls who developed weak or moderate COVID-19 (omicron).

Activated B lymphocytes (CD23) were elevated in the ozone group, probably via ozone-activated helper cells. IgG values ranged between 120 and 170 AU/mL compared with controls having 40 to 50 AU/mL (values below 10 AU/mL are considered as being a weak immune response [24].

Other trials are reported using intramuscularly administered minor autohemotherapy or ozonized saline; however, we here only refer to reports of pilot studies applying ozone systemically and applying the standardized methods of MAH and RI (Table 1 and Table 2).

Of course, more data need to be collected, and relevant parameters on redox regulation, immune status, and oxidative stress need to be measured before these promising results can be translated into a clinical concept.

### 4.2. Oxidative Preconditioning in Rheumatoid Arthritis Patients with a Second Cycle of Ozone Treatment

RA already has a history in ozone treatment. In the 1980s, we saw a complete failure; the ozone concentrations and amounts (70–80 µg/mL, volume of 200 mL corresponding to 16,000 µg ozone per 200 mL of blood in the form of MAH [37]) were too high and therefore contributed to the oxidative stress of the chronic inflammatory process. On the basis of oxidative preconditioning in redox medicine, the biochemical and pharmacological mechanisms were elaborated and our understanding of ozone as a redox bioregulatory molecule improved to a far better extent. The ozone concentration and amounts were drastically reduced in order to regulate oxidative stress instead of increasing it. Consequently, RA became one of the characteristic indications for ozone application [10,14].

Once again and renewed, the treatment of rheumatoid arthritis changes and deepens our understanding of the ozone mechanism of action: A controlled clinical trial including 20 RA patients receiving rectal ozone insufflation in 2 cycles in combination with a basic treatment (MTX (methotrexate), Ibuprofen, folic acid) and an intermediate period of 3 months [38]. The protection of liver intoxication by MTX in RA patients was reported earlier [10,14].

### 4.3. Reference Substances

Redox biomarkers—antioxidants being the protective ones and oxidative stress as injury parameters—are suitable parameters to follow the progression of the disease; the following are used to show a secondary prevention in a clinical trial on rheumatoid arthritis in a second treatment cycle where an innate immune memory by preventive ozone treatment is discussed (Table 5).

The second cycle with RI following the low-dose ozone concept and applying the same therapeutic plan as in the first treatment cycle: 1st week: 25 µg/mL, 100 mL; 2nd week: 30 µg/mL, 150 mL; 3rd week: 35 µg/mL, 200 mL; 4th week: 40 µg/mL, 200 mL, resulted in a more effective therapeutic response compared to the end of the first cycle (interval of 3 months between the end of the 1st, beginning of the 2nd cycle). Six out of seven biomarkers returned to normal; Figure 4 displays a number of them. Injury biomarkers TH and MDA reached normal values at the end of the 2nd cycle, a remarkable difference to the end of the 1st one. GSH as a protection marker increased when compared with the end of the 1st cycle in 15/20 patients; 4 patients showed a slight decrease, and in 1 patient no change was measured.

### 4.4. Involvement of the Innate Immune System

Compared with the first treatment cycle, there was a further improvement of almost all parameters at the end of the 2nd treatment cycle, which means a further enhancement of the oxidative protection mechanisms, and is understandable through a possible involvement of the innate immune system. Obviously, the innate memory is stimulated by the 2nd ozone treatment after the 3-month interval, strengthening the protection of bone and cartilage against inflammatory processes as we know them from autoimmune diseases such as RA.

Auto-antibodies CCP (anti-cyclic citrullinate peptides), which specifically correlate with RA disease progression, support this aforementioned assumption: Again, we find the same overall picture, namely, compared to the first treatment series, a significant decrease after the 2nd cycle, accompanied by a substantial improvement in the clinical course. This completely new aspect of the low-dose ozone concept in secondary prevention will be further pursued and should be consolidated by including other autoimmune diseases [38].

## 5. Mechanism of Action of Medical Ozone in Preventive Medicine

From the numerous preclinical studies on protection against hepatic, renal, or pancreatic intoxication and others, as well as from clinical studies, e.g., RA (MTX intoxication) or aging processes, the biochemical mechanism of action and the pharmacological effect of systemic ozone in preventive medicine have been elaborated and confirmed [33,34,35,36].

For prevention, the following applies: Ozone with its polar molecule structure reacts preferably with the isolated double bonds of fatty acids as they are situated in large amounts in the cell membranes. According to Crigée [39], a primary ozonide is formed, and in a second step, a secondary ozonide. This is cleaved and reacts with water to form “ozone peroxide” (hydroxy hydro peroxides), which is understood as the pharmacologically active substance (Figure 5a). These small peroxides with their terminal -C(OH)OOH group are less active than ozone itself, though still oxidants [39,40]. They are immediately reduced by glutathione GSH via ionic reaction mechanisms; this reaction sends information to the nucleus to start the redox bioregulation as Figure 5b shows.

Long-chain peroxides (Figure 5c) with a central peroxide group as initiators for radical chain reactions are responsible for permanent oxidative stress; as long-lived peroxides, they form suitable parameters for the determination of oxidative stress in redox medicine.

The “ozone peroxide” behaves as a second messenger and a regulatory molecule transducing its information via the GSH reaction to nuclear factors, mainly NFkB, which is responsible for immunomodulation, among other factors, and Nrf2, which is responsible for the regulation of antioxidants. The first step seems to be an inflammatory response to the low and specific oxidative stress of “ozone peroxide” via NfkB, and in a second step, the anti-inflammatory response via Nrf2 regulating the enzymatic antioxidant system [41], e.g., GSHox and GSHred which are needed to recover the GSH balance.

Redox bioregulation will be blocked when using high peroxide concentrations and dosages; this is the situation in patients under high oxidative stress in chronic inflammatory diseases [10].

On this basis, the low-dose ozone concept was developed; it is suitable for prevention by modulating the immune response and protecting cells from oxidative stress and free radicals through a strong antioxidant system.

## 6. Discussion

While primary prevention is usually reserved for private clinics and the demand here is increasing, especially from elderly patients and those with pre-existing conditions, secondary prevention for chronic inflammatory diseases and autoimmune diseases should also be transferred to hospitals in the form of a complementary treatment concept.

Strengthening the immune system and improving the protective mechanisms of cells through redox bioregulation will help protect patients from high oxidative stress and prevent bacterial and viral infections. Redox regulation can only take place at low ozone concentrations and doses, i.e., at mild oxidative stress, similar to the important function of hydrogen peroxide in healthy organisms: low concentrations induce, high concentrations block a regulation [42,43]. The bactericidal or antiviral effect of ozone requires direct contact to kill or inactivate the microorganisms, which is not possible in blood or in vivo.

Here, completely different reaction mechanisms come into play: In systemic applications in the form of MAH or RI, only an indirect mechanism is possible. Acute infections are therefore not an indication for the therapeutic use of ozone. We see good results in the chronic stage, for which the same mechanism is discussed as for prevention.

Due to the high reactivity of ozone, the first step is probably the formation of “ozone peroxides”, which are also unstable and are immediately scavenged by the highly potent glutathione system to trigger signal transduction via appropriate nuclear factors and initiate the production of corresponding proteins. Antioxidant enzymes are (usually) upregulated and finally, the GSH/GSSG balance is recovered.

With the measurement and tracking of GSH as a very sensitive parameter, this mechanism has been consistently confirmed in a variety of preclinical studies on prevention [14,33,34,35,36]. A first clinical study in elderly patients with age-related diseases was able to slow down further progression of the aging processes [15]. The protection against corona infections and thus against COVID-19 is another step to show the effectiveness of ozone application in prevention, as shown in the three pilot studies with a total number of 330 persons. Further trials have to be conducted, and the specific and relevant redox parameter will have to be measured to prove this concept.

Secondary prevention consists of the biochemical mechanisms which are now being expanded through a deeper understanding of a number of processes in the immune system. A repeated treatment cycle in RA patients leads to a decrease in autoantibodies indicating the formation of memory cells in the preceding treatment cycle. After the 3-month interval in ozone/MTX treatment, the second exposure to ozone apparently assumes the function of stimulating the innate immune system and, bringing the memory cells onto the scene, the oxidative protection mechanisms are further strengthened and the subsequent damage is reduced. This knowledge may well open new aspects in the treatment of autoimmune diseases.

Table 6 shows the treatment proposal following the low-dose ozone concept [32]. 

## 7. Conclusions

Preventive measures should be integrated into healthcare systems. In addition to a variety of other measures, complementary medical measures certainly could have a key role to play here, which are already being provided by a number of clinics. Promising results are also now known from clinical trials in elderly patients [15] and patients suffering from RA [14,38]. These must be confirmed, and the corresponding clinical and relevant redox markers have to be determined. The evidenced influence of the innate immune system should initiate a variety of clinical trials to investigate primary and secondary prevention.

## Figures and Tables

**Figure 1 ijms-24-15747-f001:**
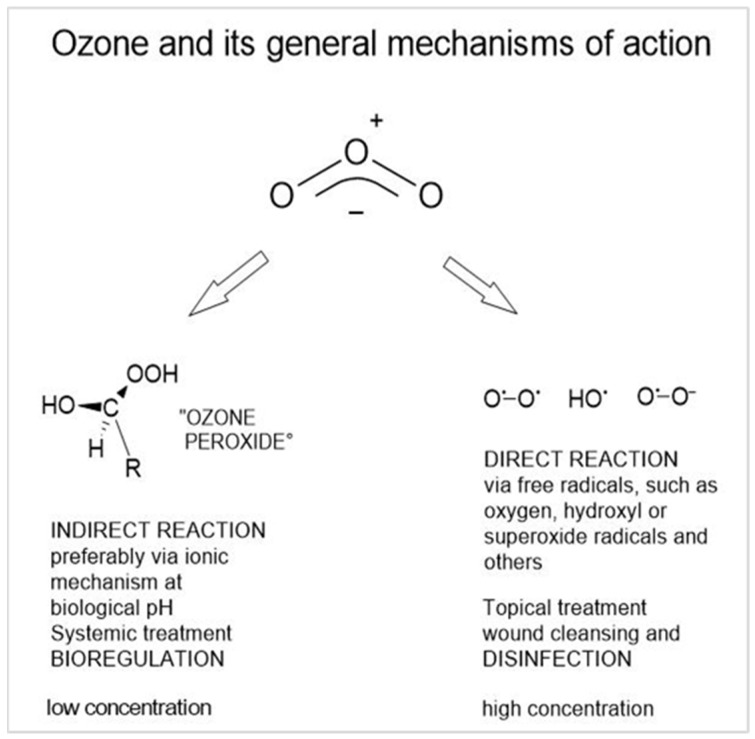
The general mechanisms of ozone depend on pH, partial pressure, temperature, on the substrate, solvents, and other factors. The ozone molecule has a polar structure, here written in its mesomeric form. Therefore, the preferred reaction is a non-radical 1.3 electrophilic addition or, at pH > 7, a radical reaction in water as solvent.

**Figure 2 ijms-24-15747-f002:**
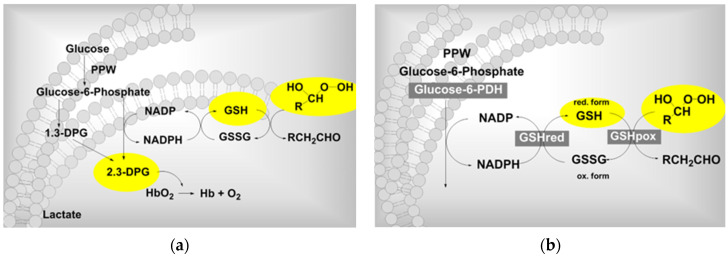
(**a**). The metabolism of RBCs influenced by peroxides, here “ozone peroxide” (formed from ozone and unsaturated fatty acids with isolated double bonds): Hexose phosphate and pentose phosphate pathways are activated at low ozone concentrations [10], whereby 2.3 DPG (diphosphoglycerate) increases. (**b**). As regards the pentose phosphate path and its function of peroxide detoxification by GSH, at high ozone concentrations (here 80 µg/mL), the GSH balance is disturbed and the antioxidant efficacy is lost [26,27]. The pentose phosphate path is induced by glucose-6-phosphate dehydrogenase providing the RBC with the antioxidant system; a deficiency is seen in patients with malaria resistance.

**Figure 3 ijms-24-15747-f003:**
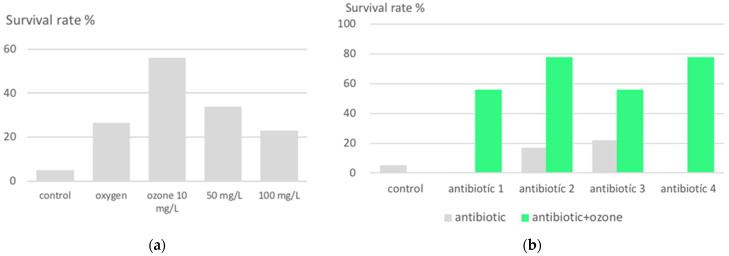
Lethal peritonitis in an animal model. (**a**) Survival rate after ozone pretreatment in dependance on ozone concentration: intraperitoneal ozone infiltration with 10 μg/mL prior to the infection shows with a survival rate of 57% as the best outcome (20 mL corresponding to a dose of 200 μg, administered 1 × daily for 5 consecutive days prior to infection); by applying an ozone concentration of 100 μg/mL, the survival rate decreases again to 23%, even worse than with oxygen. (**b**) Synergistic effects with antibiotics. Ozone pretreatment with a concentration of 10 μg/mL in all groups. Four different antibiotics were used only in one treatment, immediately after infection: Antibiotic 1: Cefodizim, antibiotic 2: Cefotaxim, antibiotic 3: Levofloxacin, antibiotic 4: Piperacillin/Tazobactam. Synergistic effects could be achieved with antibiotic 1 and 4 after one treatment only. In the cases of Piperacillin/Tazobactam (4), after a second treatment one hour later, even a 100% survival was counted [11], not shown here.

**Figure 4 ijms-24-15747-f004:**
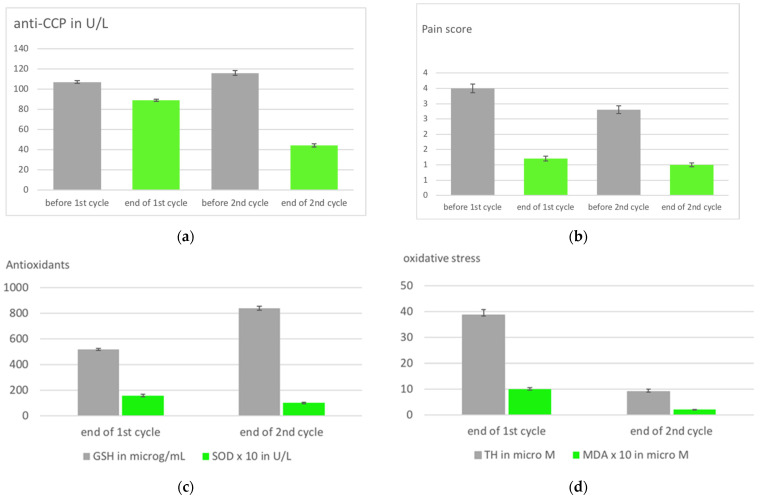
Immune memory in RA patients in a 2nd cycle of systemic ozone treatment with an interval of 3 months. Rectal insufflation in both cycles, treatment schedule (ozone concentration, volume): 1st week: 25 µg/mL, 100 mL; 2nd week: 30 µg/mL, 150 mL; 3rd week: 35 µg/mL, 200 mL; 4th week: 40 µg/mL, 200 mL. (**a**). Auto antibodies measured as anti-CCP (anti-cyclic citrullinate peptides) show a remarkable decrease at the end of the second treatment cycle corresponding to the improvement of (**b**). the clinical parameters. (**c**). Redox protection markers GSH and SOD reach normal values after the second treatment cycle (GSH normal range: 790–1100 µg/mL; SOD normal range: (50–110) × 10^−1^ U/L) (**d**). Oxidative stress measured as MDA and TH decrease according to the increase in antioxidant capacity, both are even in the reference range after the 2nd cycle (TH: 8–15 µM; MDA: (1–2) × 10^−1^ µM). Obviously, the repeated treatment is recognized by memory cells [38].

**Figure 5 ijms-24-15747-f005:**
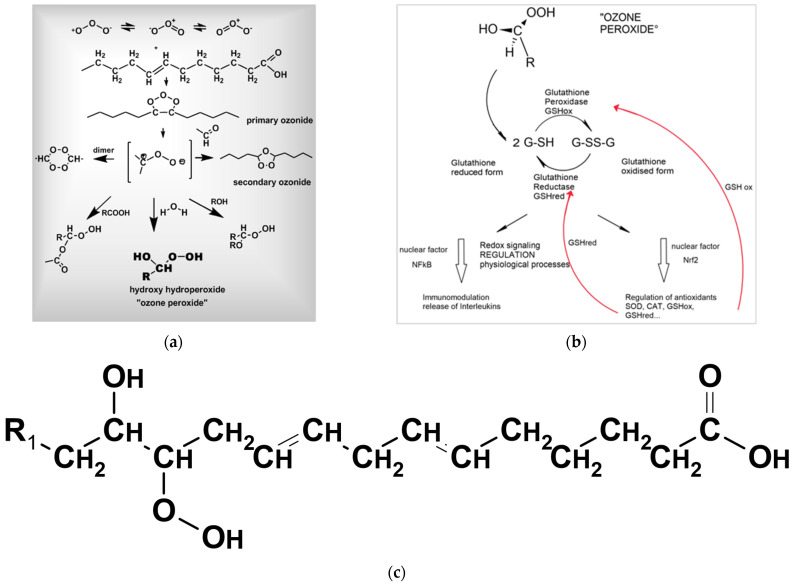
(**a**). Ozone with its polar molecule structure reacts preferably with the isolated double bonds of fatty acids as they are situated in large amounts in the cell membranes. According to Crigée [39,40], a primary ozonide is formed and secondary ozonide in the second step. It is cleaved by water and reacts to form “ozone peroxide”, which is understood as the pharmacologically active substance [10,32]. (**b**). The “ozone peroxide” behaves as a second messenger transducing its information via the GSH reaction to nuclear factors, mainly NFkB, which is responsible, among other factors, for immunomodulation, and Nrf2, which is responsible for the regulation of antioxidants, e.g., GSHox and GSHred, which are needed to recover the GSH balance (red arrows). (**c**). Long-chain peroxides: OH- and OOH-group located at different C-Atoms with the central peroxide group: lower reactivity. It is less reactive, has a longer half life, starting radical chain reactions, e.g., with oxygen. It is responsible for chronic oxidative stress: oxidative distress.

**Table 1 ijms-24-15747-t001:** Prevention from general oxidative stress. Literature survey.

Type of Study	Results	Application Route	References
Medical ozone promotes Nrf2 phosphorylation reducing oxidative stress and proinflammatory cytokines in multiple sclerosis patients.	Secondary prevention GSH increase.	RI, 20 mg/L.	Delgado-Roche, L.; Riera-Romo, M.; Mesta, F.; Hernández-Matos, Y.; Barrios, J.M.; Martínez-Sánchez, G.; Al-Dalaien, S.M. **2017**. [13].
Medical ozone increases methotrexate clinical response and improves cellular redox balance in patients with rheumatoid arthritis. Clinical trial.	Statistical significant reduction of liver toxicity in patients with rheumatoid arthritis during MTX treatment. See text.	10 rectal insufflations in 10 days.	León Fernández, O.S.; Viebahn-Haensler, R.; López Cabreja, G.; Serrano Espinosa, I.; Hernández Matos, Y.; Delgado Roche, L.; Tamargo Santos, B.; Takon Oru, G.; Polo Vega, J.C. **2016**. [14].
Medical ozone arrests oxidative damage progression and regulates vasoactive mediator levels in older patients (60–70 years) with oxidative etiology diseases. Controlled clinical study.	Positive influence on aging process. See text.	Rectal insufflation.	León Fernández, O. S., Takon Oru, G., Viebahn-Hänsler, R., López Cabreja, G., Serrano Espinosa, I., García Fernández, E. **2022**. [15].
Modulation of Oxidative Stress by Ozone Therapy in the Prevention and Treatment of Chemotherapy-Induced Toxicity: Review and Prospects.	Prevention from toxicity, mainly in animal models.	Systemic applications.	Clavo, B.; Rodríguez-Esparragó, F.; Rodríguez-Abreu, D.; Llontop, P.; Aguiar-Bujanda, D.; Santana-Rodríguez, N. **2019**. [16].

**Table 2 ijms-24-15747-t002:** Literature Survey.

Type of Study	Results	Application Route	References
Ozone Therapy for Prevention and Treatment of COVID-19. Review.	4 publications on prevention.	Different forms of application.	Martínez-Sánchez, G. **2022**. [17].
Intravenous ozonized saline therapy as prophylaxis for healthcare workers (HCWs) in a dedicated COVID-19 hospital in India. A retrospective study.	Less infections in the ozone group (4.6%) than in the control (14.03%). n = 64 (235).	Ozonized saline.	Sharma, A.; Shah, M.; Sane, H.; Gokulchandran, N.; Paranjape, A.; Khubchandani, P.; Captain, J.; Shirke, S.; Kulkarni, P. **2021**. [18].
Immunity Prophylaxis with Ozone Therapy. Review Report.	2.19% incidence rate (n = 320).	Minor autohemotherapy.	Shah, M.; Captain, J.; Ganu, G. **2020**. [19].
Could the minor autohemotherapy be a complementary therapy for healthcare professionals to prevent COVID-19 infection?	(n = 73) No infection or positive test.	Minor autohemotherapy.	Orscelik, A.; Karaaslan, B.; Agiragac, B.; Solmaz, I.; Parpucu, M. **2021**. [20].
The Role of Ozone Therapy as Adjuvant in the Management of COVID-19 in Indonesia. Clinical trial.	See text.	MAH, RI, and vaginal insufflation.	Chaijadi, D.; Hendradiana, A.; Tjahjono, P.D.; Kusumaningsih, E.; Hariyanto.; Siagian, C.; Atmadja, D.S.; Viebahn-Hänsler, R. **2023**. [21].
Could ozone therapy be used to prevent COVID-19? Clinical trial.	2 of 71 persons were tested positive. Retrospective, no control. 45% medical professionals. See text.	MAH, 10 treatments.	Gencer-Aalay, K.; Sahin, T. **2022**. [22].
COVID-19 prophylaxis with ozone therapy.	n = 9, good effect but no protection from further infection.	RI.	Falzoni, W.; Senvaitis, M.I.; Iwasa, S. **2021**. [23].
Comparative analysis of 2 groups of people according to age and sex, vaccinated triple versus COVID-19, were subjected to quantitative analysis of antibodies and B lymphocytes after ozone therapy. Clinical trial.	See text.	MAH.	Medina, J.G. **2022**. [24].

**Table 3 ijms-24-15747-t003:** Concentrations and dosages in systemic ozone treatments following the low-dose ozone concept [32].

	MAH	RI	Main Indications
Ozone concentration	10 to 40 μg/mL	10 to 40 μg/mL	Chronic inflammatory diseases or diseases accompanied by inflammation, such as RA, angiopathy. Age-related diseases, chronic intestinal diseases. Prevention.
Volume	50 to 100 mL	150 to 300 mL	
Dose	500 to 4000 μg	1500 to 12,000 μg	

**Table 4 ijms-24-15747-t004:** Clinical trials in COVID-19 antibody production. Patients and treatment procedures.

Control Group, n = 52	Ozone Group, n = 57
3 × vaccinated: n = 33 (63.5%); 2 × vaccinated: n = 19 (36.5%)	3 × vaccinated: n = 31 (54.4%); 2 × vaccinated: n = 26 (45.6%)
female n = 33, male n = 19 age 20 to 55	female n = 24, male n = 33 age 20 to 55
	MAH: 1 × per week, 8 treatments RI: 2 × per week during 2 weeks, then 1 × per week; 15–16 treatments

**Table 5 ijms-24-15747-t005:** Characteristic injury and protective redox markers and some clinical parameters in RA patients during a two-cycle systemic ozone treatment, see Figure 4.

Injury Biomarkers: Oxidative Stress Markers	Protective Markers: Antioxidants	Clinical Parameters
TH (total hydroperoxides)	Total SOD (superoxide dismutase) CAT (catalase)	DAS (disease activity score) PAIN
MDA (malondialdehyde) AOOP (advanced oxidation protein products)	GSH (reduced glutathione)	HAQ: DI (health assessment disability questionnaire)
NO (nitrogen monoxide)	GGT (gamma glutamyl transferase)	Auto antibodies CCP (anti-cyclic citrullinated peptides)

**Table 6 ijms-24-15747-t006:** Ozone and prevention: treatment recommendation.

Application	Ozone Concentration	Ozone Volume	Treatment Frequency
RI	15–25 µg/mL In general: 10 to max. 40 µg/mL	150–300 mL	twice per week, if possible prior to chemotherapy or antibiotics. At least once per week as adjuvant therapy two or 3 series per year with 10 treatments each.
MAH	15–25 µg/mL In general: 10 to max. 40 µg/mL	50 mL	2 or 3 treatments per week, if possible prior to chemotherapy or antibiotics. At least once per week as adjuvant therapy or 2 to 3 series per year with 10 treatments each.

## Data Availability

Data are contained within the article.

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
