# Peer review of "Ozone as Redox Bioregulator in Preventive Medicine: The Molecular and Pharmacological Basis of the Low-Dose Ozone Concept—A Review"

_ijms, 2023, doi:10.3390/ijms242115747_

Round 1

Reviewer 1 Report

In this manuscript, Viebahn-Haensler and León Fernández reviewed the recent literature on ozone administration as a preventive treatment in the clinical practice: so far, this has been seldom investigated and proposed as therapeutic approach, thus the manuscript is potentially interesting and worth of considering for publication.

However, in its present form, this paper cannot be accepted and needs revision.

·      First of all, since literature findings are recalled and discussed, it would be more appropriate to arrange it as a review than as an original article.

·      In the text, the low-dose concept is often cited but not explicitly defined (in the Abstract it is even not mentioned): to make the Title fully consistent with the article content, the authors should clearly explain the low-dose concept the first time it is recalled.

·      Lines 62 and 113: “Schulz” should be “Schulz et al.”

·      Line 76: the heading “Results” should be deleted as it is inappropriate (actually, data from the literature were only recalled and no original findings). The following sections/subsections must be properly renumbered.

·      Line 79: “Plasmodium falciparum” in italics.

·      Line 88: “Mechanism of action” may be deleted.

·      Lines 99-100: this sentence is rather obscure to readers who are not aware of the regulatory role of ozone in the eustress induction: please rephrase with appropriate references.

·      Line 122: references should be quoted (e.g., no. 10, 12 and 20).

·      Line 134: “as” should likely be “is”.

·      Line 139: “In the late 1990´s León started a broad-based research project on ozone-oxidative preconditioning”, but no reference is quoted: please complete.

·      Line 146-8: sentence unclear to be rephrased.

·      Lines 164, 170, 178, 234: “Procedure” and “Results” are useless and may be deleted.

·      Lines 206-10: appropriate references should be added

·      Lines 214-6 and lines 223-6: sentences unclear to be rephrased.

·      Lines 206-32: This paragraph on the mechanism of action of medical ozone in preventive medicine should probably better be moved before the section Discussion

·      Line 313: the section title “Material and methods” is inappropriate in this review article and should be deleted; however, since the paragraph al lines 314-22 clearly describes the criteria the authors followed in their survey, it may be moved after line 148.

In the main text, evidence of some carelessness was found:

·     Abbreviations must be spelled put the first time they are used, and when an abbreviation is defined, then this must always be used (see e.g, rheumatoid arthritis that was repeatedly used after RA was defined at line 234, although it first appeared at line 73).

·     Milliliter must be “mL” throughout.

·     Both “ozone” and “O3” were used: please, choose either one.

·      Throughout the text, care must be given to the proper use of the fonts and font sizes.

Mistyping and misspellings are present (e.g., lines 134 and 237) while some sentences are incomplete, so that they actually are meaningless (e.g., lines 52-3; 111-13; 141-3).

A revision by a native English-speaking expert is recommended.

Reviewer 2 Report

Dear authors,

your work on ozone as a redox signal molecule is very relevant. However, in its current form, the paper needs significant improvement. While indicating that your interest is low dose treatment and the mechanisms behind it, many citations and much of the text is dedicated to the well-accepted (chemical) disinfection by ozone (and AOPs in a wider sense). The more recent, interesting parts, e.g. the ozone peroxide, lack sufficient support from literature and discussion. To my mind, the paper should have a clearer focus and a much-extended discussion on the potential mechanisms behind the different observations you picked from the literature.

Please also see the following remarks:
general:

- language editing required

- organization of the paper

- scope of the paper remains unclear - it shifts between the well established use for disinfection purposes and emerging, in vitro-research driven experiments on redox biology manipulation by ozone 

- the manuscript remains too descriptive 

- literature basis too weak for this type of paper

abstract

- meandering between "chemical" disinfection (including CoVID...) by ozone and biological responses to ozone by the immune system - which are fully different aspects and mechanisms

- missing hint that this is an opinion paper (or a kind of meta-study)

introduction

- fig. 1 proposed reactions (especially the addition to form a geminal mixed ol/hydroperoxide) need citations to underpin their relevance; this mechanism of reaction is referred to later in the manuscript without sufficient proof (line 211ff)

- in my opinion, the reaction of ozone with water (or H2O2) forming subsequently active radicals is neglected in this work

results

- the selection of past studies looked into in detail seems somewhat arbitrary

- study on P. falciparum - it remains unclear to the reader how the study was performed, and the proposed mechanism focuses on RBC (which in the study did not respond in the expected way), leaving out F. falciparum treatment that was effective

- study on intraperitoneal injection - no biochemical mechanisms discussed, the data of this study are only reproduced without further processing

- prevention of COVID: only one of the selected studies used controls, and no information on the biochemical mechanisms was collected; the discussion is not well organized and does not focus on the reported studies 

- study on rheumatoid arthritis: no controls, no crossover - ozone is administered as an add-on to normal therapy that hampers detailed conclusions on the role of ozone

discussion

- extent and scope not sufficient

Kind regards

editing required, see above
